# RFID Applications and Security Review

Cesar Munoz-Ausecha * , Juan Ruiz-Rosero and Gustavo Ramirez-Gonzalez

Department of Telematic Engineering, Universidad del Cauca, Popa 190002, Colombia; jpabloruiz@unicauca.edu.co (J.R.-R.); gramirez@unicauca.edu.co (G.R.-G.)
* Correspondence: munozausecha@unicauca.edu.co

**Abstract:** Radio frequency identification (RFID) is widely used in several contexts, such as logistics, supply chains, asset tracking, and health, among others, therefore drawing the attention of many researchers. This paper presents a review of the most cited topics regarding RFID focused on applications, security, and privacy. A total of 62,685 records were downloaded from the Web of Science (WoS) and Scopus core databases and processed, reconciling the datasets to remove duplicates, resulting in 40,677 unique elements. Fundamental indicators were extracted and are presented, such as the citation number, average growth rate, and average number of documents per year. We extracted the top topics and reviewed the relevant indicators using a free Python tool, ScientoPy. The results are discussed in the following sections: the first is the Applications Section, whose subsections are the Internet of Things (IoT), Supply Chain Management, Localization, Traceability, Logistics, Ubiquitous Computing, Healthcare, and Access Control; the second is the Security and Privacy section, whose subsections are Authentication, Privacy, and Ownership Transfer; finally, we present the Discussion section. This paper intends to provide the reader with a global view of the current status of trending RFID topics and present different analyses from different perspectives depending on motivations or background.

**Keywords:** RFID; radio frequency identification; scientometric; bibliometrics; data science; ScientoPy

## 1. Introduction

It is expected that technology will work for us in more transparent and ubiquitous ways. RFID is implemented in many contexts, such as logistics [1,2], supply chains [3,4], asset tracking [5–7], healthcare [8,9], industrial enterprise environments [5,6], and many others, making our lives easier and generating vast quantities of information on many levels.

This review paper describes the most active topics of investigation regarding RFID technology. This information will help researchers in identifying research gaps and trending topics. Additionally, it could be useful for other disciplines and those outside academia, providing a brief overview of the most active investigations into RFID applications and security topics.

Multiple reviews and surveys exploring RFID from various perspectives have been published. From 2007 to 2009, there were reviews on the applications of this technology [10,11], as well as academic literature and historical reviews [12,13]. Additionally, we encountered a review published in 2010 on how RFID is used for tracking in the Internet of Things (IoT) [14]. A review of specific applications such as in postal and courier services was published in 2006 [15]. In the same way, work on its use in the aviation industry was published in 2010 [16]; work on its use in activities such as construction was published in 2015 [17]; there were healthcare-related publications from 2013 to 2018 [18–20]; work on supply chain management was published between 2010 and 2016 [21,22]; work on localization in the smart home paradigm was published in 2018 [23]; finally, in 2020, a systematic literature review was published on the applications of RFID in supply chains and its impacts on the competitive advantages of organizations [24], analyzed from corporate,

customer, and benefit perspectives. There was also a review of the role of RFID in transportation [25], identifying the advantages of, and barriers to, its adoption. We observed that these reviews tended to focus on one specific topic each year.

This review compiles the most active topics to present the trends in the applications of RFID technology, based on the numbers of citations in papers published from 1996 to 2020 and the activity in the selected topics between 2019 and 2020, contributing an updated overview similar to the reviews of the applications published before [10,11], but also creating eight groups based on the most active topics from 2008 to 2009 [12,13], complementing this with the security, privacy, and ownership transfer topics.

Initially, we describe the process of working with separate datasets and reconciliation to obtain unique dataset entries, and we present the indicators related to the datasets used. We start with the RFID Applications Section, followed by the Security and Privacy section. The first section is subdivided based on the most active subtopics encountered in the analysis. The subsections discuss the Internet of Things (IoT), supply chain management (SCM), applications for localization using RFID, traceability, logistics management, ubiquitous computing, RFID in healthcare, and applications in access control. In the second section, we present the leading security and privacy topics, and in the subsections contained therein, we present authentication and ownership transfer. Finally, we present the Discussion.

## 2. Materials and Methods

This article collected and used information from the Scopus and Clarivate Web of Science (WoS) databases. To download the raw source datasets, an institutional subscription was needed to access and retrieve the data entries from both databases. Using the requisite web applications to access the databases, we performed the query using the string ("Radio frequency identification" OR "Radio-frequency identification" OR "RFID") to obtain the data registered in each database. The query was executed on 4 February 2021. Subsequently, the data were processed using the scientometric tool ScientoPy [26].

The process applied to reconcile the datasets obtained from both databases, WoS and Scopus, and to generate the secondary dataset is represented in Figure 1. For more details, see the first three sections of the ScientoPy manual available in the GitHub repository https://github.com/jpruiz84/ScientoPy (accessed on 7 February 2021).

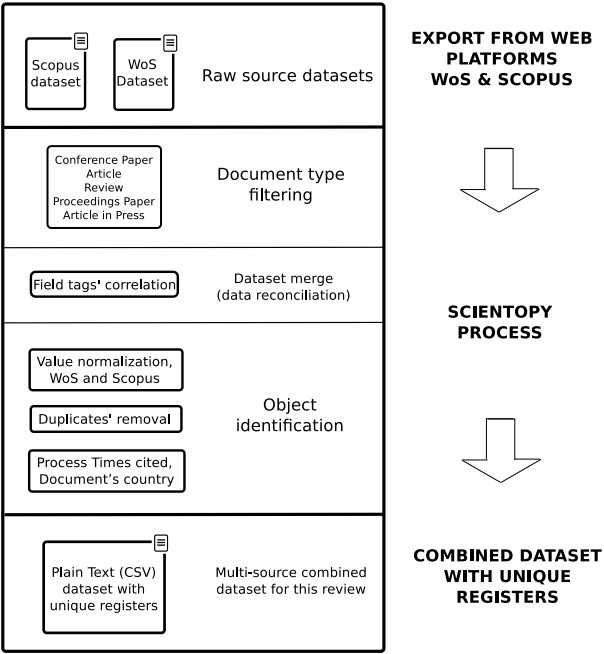

**Figure 1.** Data process to remove duplicates and obtain unique entries from the WoS and Scopus datasets, using ScientoPy [26].

*Data Description*

The raw source dataset used for this work was retrieved on 4 February 2021. It contains 62,685 papers, appending entries from the WoS and Scopus databases. After the data reconciliation [27], we worked with 40,677 individual entries from both databases containing 24,587 unique papers from WoS and 16,090 unique papers from Scopus, removing 360 duplicated papers from WoS and 21,648 from Scopus. The distribution is shown in Figure 2.

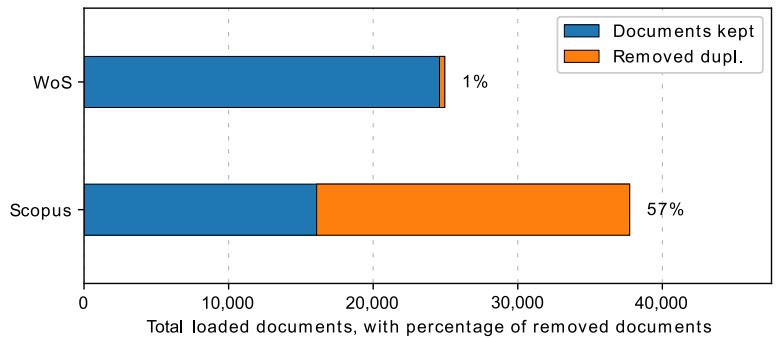

**Figure 2.** ScientoPy data preprocessing from datasets downloaded from Clarivate Web of Science and Scopus, showing unique papers in blue and removed duplicates in orange.

Table 1 defines the acronyms for the statistics determined by ScientoPy; the software uses three different topic growth indicators to find trending topics, including relative/absolute growth, and calculates the h-index of each topic.

**Table 1.** Topic growth indicators.

| Indicator | Complete Name | Description |
|-----------|---------------|-------------|
| AGR | Average growth rate | The average difference between the number of documents published in one year and the number of documents published in the previous year. It indicates how the number of documents published for a topic has grown (positive number) or declined (negative number) on average in a time frame. |
| ADY | Average documents per year | This is an absolute indicator that represents the average number of documents published inside a time frame for a specific topic. |
| PDLY | Percentage of documents in last years | Relative indicator that represents the percentage of the ADY in relation to the total number of documents for a specific topic. |
| h-index | Author-level metric | Using the "times cited" field, ScientoPy calculates the h-index of each topic for the different categories, such as authors, countries, institutions, and others. |

## 3. RFID Applications

RFID technology is making its way into various fields, sometimes keeping a low profile. However, it is becoming more relevant in people's lives, creating paths to new applications and complementing known ones. In some cases, these applications do not directly interact with people, but RFID supports different processes for realizing products and services used daily. Accordingly, with reason, RFID is sometimes seen as a requirement or core technology for the Internet of things (IoT). This is affirmed by the total documents published on this topic, positioning IoT well above other applications. RFID supports many applications of the Internet of Things due to characteristics such as high volume,

non-contact reading, tag features and, more importantly, low price. The situation is clearly reflected in the graphs (a) and (b) in Figure 3, with the more prominent indicators in Table 2. In second place, we have supply chain management, in which RFID is essential for identification and information capture for real-time management, using different methods such as frameworks and protocols. This allows high versatility due to the variety of tags available; all these processes are continually studied to solve security issues and possible vulnerabilities. RFID is also used in localization applications, extending the primary function of the technology, representing the second-largest activity in terms of the percentage of documents in last years (PDLY), as shown in Table 2. For traceability and logistics, RFID supports a variety of applications in supply chains and shop floor deployment. RFID is opening paths to new applications in an ever more ubiquitous fashion, blending into our lives much more, such as in healthcare spaces for ensuring data security and allowing more robust management of patients, equipment, and personnel. Similarly, it is applied to support the control of access to physical spaces and systems or enforcing the control of access to information captured by different sensors along with RFID. All these findings are presented in detail in the following sections and represented graphically in Figure 3. This chapter presents the top research applications identified through the scientometric analysis.

**Table 2.** Top author keywords for RFID applications.

| Pos | Author Keywords | Total | AGR | ADY | PDLY | h-Index |
|-----|-----------------|-------|-----|-----|------|---------|
| 1 | Internet of Things | 2399 | 8.0 | 348.0 | 29.0 | 73 |
| 2 | Supply Chain Management | 408 | 1.0 | 16.5 | 8.1 | 49 |
| 3 | Localization | 402 | −4.5 | 28.0 | 13.9 | 32 |
| 4 | Traceability | 307 | −1.5 | 17.0 | 11.1 | 29 |
| 5 | Logistics | 264 | −1.5 | 11.0 | 8.3 | 20 |
| 6 | Ubiquitous Computing | 210 | 0.5 | 4.0 | 3.8 | 25 |
| 7 | Healthcare | 165 | 1.0 | 11.0 | 13.3 | 23 |
| 8 | Access Control | 131 | 2.0 | 8.0 | 12.2 | 14 |

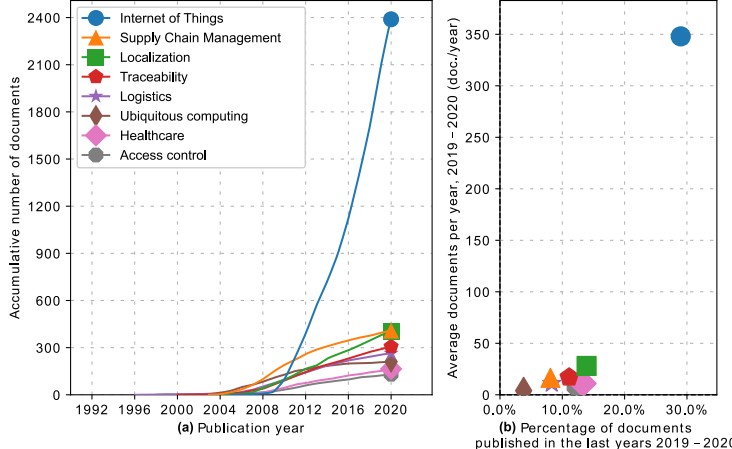

**Figure 3.** (**a**) Top applications, and number of publications by year. (**b**) Average number of publications per year and percentage of documents published between 2019 and 2020.

### 3.1. Internet of Things (IoT)

With RFID, we can track and securely monitor individual objects worldwide in different environments. The technology offers multiple cost options, depending on the conditions and requirements. For this reason and more, RFID is often seen as a prerequisite for the IoT [28–30]. There are multiple studies on communication protocols to improve performance without compromising on security and scalability [31]; strong security and privacy are important when using mutual authentication between the reader and the tags [32]. The

existing security attacks, privacy risks, and countermeasures to be applied [33], and identifying the limitations, requirements, and issues to work on for more secure deployment to meet security demands [34–37] are also being studied. However, there is a demand to search for more lightweight RFID authentication schemes due to the limitations of processing, storage, and power on the RFID tag side [38,39]. Security and privacy communication models are being developed that are better optimized for limited resources, while meeting the security requirements of applications [34,40]. Additionally, investigations are focusing on commercial tags to identify common flaws and possible measures to realize secure applications [41,42]. RFID–IoT investigations are equally analyzing how to enforce security through new policies for fine granularity and context-aware information control [43]. Additionally, to test these new proposals, there is a hardware and software suite for laboratories for emulating tags and readers [44] and analyzing security schemes that aim to provide secure and untraceable communication for end-users [45], allowing flexibility and application in different environments, considering important IoT components such as quality of service (QoS) or message queue telemetry transport (MQTT) [46,47].

RFID plays a role in the industrial enterprise environment, taking into consideration the low-carbon economy [48]. However, its main role is helping in monitoring, traceability and tracking [5], or obtaining more accurate location information by using ultra-wideband (UWB) technology [6]. It also helps to ensure the safety of personnel. For instance, by using wireless sensor networks (WSN) along with RFID, researchers aim to implement an advanced safety system for the hazardous environments in industrial plants [49]. The industrial Internet of Things (IIOT) requires a high level of trust and performance due to the significance of these tasks. Hence, there have been investigations into a general architecture for application program interfaces (APIs) with which to handle the considerable diversity of RFID equipment available for industrial environments [50], and to improve communication speed and reliability for a large population of tags using an estimator to maximize the efficiency of channel usage [51].

RFID communication protocols offer protection against tag tracking, replay attacks, and cloning attacks, among others, enabling a balance between security and high efficiency [52,53]. Alternatively, some protocols tend to be lightweight or ultra-lightweight, such as the mutual authentication protocol, revised to be more secure and enforced using a cache on the reader side [54,55]. However, investigations are going further and defining an address translation protocol enabling RFID to IP for complete IoT communication [56]. Equally, for security, cryptanalysis is applied to ultra-lightweight mutual authentication to ensure untraceability and identify vulnerabilities [57–59]. Another concern in RFID protocols is privacy, which, in some cases, limits the use of this technology. Therefore, investigators are working on an anonymous authentication protocol to protect against clandestine tracking [60,61]. Consequently, to test and validate these changes, it is necessary to use a framework to model and simulate new protocols to evaluate their robustness [62].

### 3.2. Supply Chain Management (SCM)

RFID for supply chain management (SCM) plays a critical role in connecting the physical world with management systems; RFID collects data to be analyzed and improves the quality of management, which is an area of special interest for large enterprises [63]. For example, by applying a framework for the development of RFID, the operation efficiency can be improved, and competitive advantage can be enhanced [64]. Additionally, in the search for a smarter process [65], researchers are analyzing the effect in e-supply chains for grocery retailing [66]; the benefits, challenges, and effects that RFID applications may bring for retail in the UK [67]; the application in convenience stores in Taiwan [68]; and comparing RFID benefits for USA and Republic of Korea (South Korea) [69]. However, it is important to perform a mean-variance analysis to implement smart shelves [70], with identification at the item level in the retail supply chain, and determine how the cost should be allocated among the partners and its application on a sales floor [3,4]. In the same way, the available software solutions and information systems for supply chain coordination have

been investigated [71], assessing the performance gained by implementing RFID [72,73], using methods such as quantitative evaluation [74] or using empirical evidence [75,76]. By contrast, investigators consider different variables in the supply chain, such as the effects of tag orientation and package contents affecting the tag reading, resulting in inconsistencies [77]. There is a drive for increased speed in processes using RFID portable devices in construction supply chains [78]. The supply chain can generate a vast amount of data that need to be processed and managed [79] using an efficient storage scheme and query process on relational databases [80,81], no-relational repository designs in MongoDB [82], or a method for processing massive amounts of data using path encoding [83]. All these data can be either important or not, based on their attributes [84]. For this reason, it is necessary to use a model to quantify the quality of the data [85], according to factors that may affect the economy. Inventory register inaccuracies can have negative consequences [86,87]; on the other hand, the adoption of RFID can help to prevent counterfeiting and fake products [88,89], in critical supply chains such as medicine [90–92], where the protection of privacy and security is of utmost importance [93–95]. These advantages acquired by using RFID come with economical costs to be considered [96–98]. Finally, RFID can be applied in close-loop supply-chains to cut costs in large business applications [99–101].

### 3.3. Localization

RFID technology's primary function is the identification of objects, but investigations are pushing beyond the limits of this technology to expand its capabilities, allowing localization information to be obtained from specific indoor scenarios where global positioning system (GPS) devices generally do not work. Examples of these applications are real-time location tracking with high precision via RFID tags using commercial off-the-shelf (COTS) devices [7], a standalone positioning system using low-cost passive RFID tags installed to guide the path [102], the location of tags that are not in line of sight (NLOS) for the reader [103–105], and navigation for autonomous mobile robots based on passive RFID tags to be applied in indoor environments [106–108]. Alternatively, for industrial mobile vehicles and robots, the signal strength indicator from the vehicle's transponder makes the vehicle move when RFID UHF tags are read [109,110]. In a similar way, it enables the location and approach to a stationary target object using the same principle [111]. Another example is occupancy monitoring for smart buildings and indoor environments, focusing on improving the energy usage in these spaces [112,113]. RFID is even used in the construction of buildings, enforcing management and safety, besides facilitating the location and tracking of people, equipment, and materials in realistic construction environments [114–116]. Additionally, RFID is used to enable high accuracy in assisted positioning in vehicular ad hoc networks (VANETS) [117]. The use of ultra-wideband (UWB) is ever advancing, overcoming most of the current limitations of narrowband, such as low area coverage, interference, and limited multiple-access capability [118]. This is enhancing the location applications of RFID, and new propagation models are being tested [119], with alternative location techniques for tags using phase differences, obtaining better accuracy, robustness and sensitivity when integrated with other measurements [120]. Consequently, investigators are finding solutions to this technology's challenges, such as through millimeter-level ranging based on backscatter RF tags [121] or the usage of phased array antennas to determine the precise angles and positions of small passive RFID tags [122].

### 3.4. Traceability

Traceability provides information about the event history, locations, or application of tagged elements in the supply chain, including origin and destination. Traceability systems based on RFID are being studied, aiming at a reduction in costs, improving customer service and decision making [72], and allowing different applications, such as traceability in food supply chains [123]; traceability in the context of cattle/beef quality and safety in China [124]; traceability systems incorporating 2D bar codes and RFID for wheat flour

mills [125]; the use of RFID, bar codes, and digital weighing technologies in manual fruit harvesting [126]; advanced tracing in aquaculture supply chains, allowing the product to be traced from the farm to the consumer [127]; and performance evaluation. This is relevant not only in intercontinental flesh fish logistics [128], but also in the supply chains of living fish [129]. Lessons have been learned from its application in an aircraft engineering company [130], and the tracing of beef from farm to slaughter in Ireland [131]. Moreover, there have been advances in data analysis using semantic trajectories centering on the behavioral patterns of moving objects, without neglecting the privacy issues that arise due to the semantic aspects of trajectories [132]. Realizing RFID-based traceability improves production scheduling in a job-shop environment [133]; traceability across sovereign, distributed RFID databases [134]; and the automatic tracing of production with mobile and agent-based computing [135]. Moreover, it is important to know the perceived benefits and drawbacks of the technologies used in tracking to identify the factors that influence consumers' perceptions of these technologies in their lives [136].

### 3.5. Logistics Management

In logistics, RFID technology is used in acquiring data that allow the visualization and production of analytics, such as on a shop floor, where large quantities of data are generated [1,2], and material tracking for construction logistics [137]. Applications such as these demand the implementation of intelligent network systems for collecting information to be analyzed and deployed in real time [138]. Moreover, studies on the retail sector have led to a contingency model for identifying the most important elements of RFID supply chain projects for application in logistics and manufacturing environments [139]. Similarly, researchers have defined a framework for assessing RFID applications for logistics [140]. There is research on optimizing operations using a distributed simulation platform to design advanced RFID-based freight transportation systems [141]. This will further allow vehicle management in logistics to work along with global positioning systems (GPS) and geographic information systems (GIS) [142]. An architecture for improving terrestrial logistics has also been described [143]. Additionally, researchers have worked on a method based on fuzzy scorecard for logistics management in robust supply chain management (SCM) to evaluate the overall performance in real-time [144]. An analysis and comparison of RFID and Auto-ID for logistics performance were also conducted [145]. Thus, it is possible to extend the functionality in logistics using the simultaneous operation of RFID and bar-codes [146], or the knowledge-based logistics management system (KLMS) design to support service providers in making more contextualized decisions [147].

### 3.6. Ubiquitous Computing

As mentioned previously and shown in the previous and following sections in this paper, RFID is present in many places and environments. It usually works in a transparent manner. This section describes different examples and utilities for ubiquitous RFID— for example, tracking autonomous entities in indoor environments using RFID, and makes a comparison with other technologies [148]. RFID is ubiquitous by its use, not only in home-health monitoring with context awareness [149], but also in mobile telemedicine systems for perceiving emergencies in older adults [150]. RFID is also applied in the classroom for identifying and obtaining context services, focusing on natural interaction with the system [151]. Alternatively, in large deployments such as the smart cities case, it helps in identification in massive streams of real-time data [152]. On the other hand, it can be present even in a simple game of cards, adding new features using computers [153]. This type of application makes the creation of frameworks to support the process of creation necessary, making the development easier, as shown in [154]. It can be seen that this technology is advancing in every regard; whether this is a good situation can be questioned, considering the impacts of ubiquitous computing from legal and social perspectives [155].

### 3.7. Healthcare

RFID is important in the Internet of Things (IoT) ecosystem. One can observe the active integration of RFID in healthcare environments, supporting various processes. This section describes some of the solutions implemented or explored for these sensitive environments. For example, in IoT-based healthcare smart spaces, investigators have explored the use of RFID in body-centric systems and for gathering information such as temperature, humidity, and other variables related to the user's living environment [8]. In addition, RFID is used in devices for managing diabetes therapy in ambient assisted living (AAL) spaces [9]. It is also used to store healthcare events on large HF RFID memory tags and integrated with Twitter services to keep the patient's relatives up to date [156]. The IoT is applied to healthcare through RFID-enabled asset management to evaluate the advantages and impact obtained [157,158]; there is an IoT hybrid monitoring system for healthcare environments called "CUIDATS", which integrates RFID and wireless sensor network (WSN) technologies [159]. RFID-IPv6 has been used in healthcare, making the deployment less costly with less maintenance [160].RFID is applied for protecting medical privacy to mitigate the risk of the disclosure of patient data, along with an IoT security infrastructure [161]; an RFID-based security platform for a healthcare environment where the data collected from the patient is extremely sensitive has been developed [162]. A signing platform based on RFID to manage health workers regarding work efficiency in real time has been realized [163]. Other applications can be found in an intelligent warning system to provide real-time visual and auditive warnings to other drivers when an emergency vehicle needs to pass through at an intersection [158].

### 3.8. Access Control

Access control can limit entry to a physical space or access a tag's data or backend services through a middle-ware system; as a result, RFID is applied in different ways to control multiple types of systems. One application is protecting people's privacy by excluding irrelevant information for spaces under video surveillance, by limiting the recording based on RFID [164,165]. This access control can be performed by applying frameworks to stop unauthorized readers from accessing tags that could result in unauthorized access to stored information [166]. In the same way, when the access needs to be limited to the right stakeholders and systems, different methods have been investigated, for example, asymmetric keys for mutual authentication [167], digital signature systems [168], role-based access applied to the readers [169], or using protocols based on the ownership of the tags [170]. In the past, systems have been breached, such as the case of the cloning of MiFare Passes [171]. Therefore, practices such as the reuse of tags by multiple companies can be dangerous [172]. This necessitates the analysis of the security used and fine credential management [173] for data access control [174,175] or document access control [176]. However, in mobile objects, the implementation of security significantly degrades the system's performance or changes the data capture, affecting the storage of information, which has led to the development of models and policies for improving confidentiality protection and governance for object tracking data [177,178]. There are access control methods based on IPv6 [179], more simple approaches that work without a backend system [180], and hybrid systems that support online and offline modes, to adapt according to the rules [181]. These methods are used, for example, to control access for vehicles [182] and healthcare environments [182]. Equally important are management systems for applying access control and positioning security [183]. The systems offer different authentication and access control methods that need to be analyzed to guarantee their safe application in different environments [184]. In the same way, the GS1 Electronic Product Code Global (EPCglobal) network is designed to provide a common language for supply chains by which to share information in a secure manner [185]. However, new models are sometimes necessary to maintain the level of privacy in the process [186], similarly to the concerns regarding privacy and access control policy implementation to control middle-ware systems [187,188]. Finally, it is observed that investigations tend to focus on futuristic applications such as human microchip implants,

making necessary the evaluation of the relationship between risks and rewards [189], analyzing if there exists a more secure method of implementation [190], and exploring the current perceived barriers in these application scenarios [191].

## 4. Security and Privacy

One of the major concerns in many applications is the security and privacy of data. The main factor is the fast expansion and Internet diversification in the IoT and related services. This has led to various concerns, such as those regarding the security of currently used protocols, such as the mutual authentication protocol. Additionally, it is necessary to analyze existing attacks and their solutions, and evaluate currently used standards such as the EPC Global C1G2, to better protect the main technology users.

Investigations are testing and searching for new methods for RFID readers, tags, middle-ware, and backend systems. The relevance of this situation can be confirmed by the average number of publications per year related to security shown in Figure 4, with the more important indicators shown in Table 3, presenting values close to the IoT topic in the application section in Table 2. Thus, the authentication topic shows the same behavior. The majority of the studies are focused on the authentication between the reader and tags, exploring new methods for protecting this communication against different identified attacks, maintaining a steady AGR as presented in Table 3. Similarly, the ownership transfer topic explores new methods for enforcing security in the actual transfer of tags, protecting the privacy of the parts and exploring the scalability of the process to secondhand markets and circular supply chains. The evolution of these topics is shown in graph (a) on the left in Figure 4, and the topics are presented in more detail below.

**Table 3.** Top author keywords for security- and privacy-related topics.

| Pos | Author Keywords | Total | AGR | ADY | PDLY | h-Index |
|-----|-----------------|-------|-----|-----|------|---------|
| 1 | Security | 1974 | 2.0 | 107.0 | 10.8 | 57 |
| 2 | Authentication | 692 | 0.5 | 42.0 | 12.1 | 44 |
| 3 | Privacy | 671 | −2.0 | 19.0 | 5.7 | 49 |
| 4 | Ownership transfer | 93 | −2.0 | 3.5 | 7.5 | 16 |

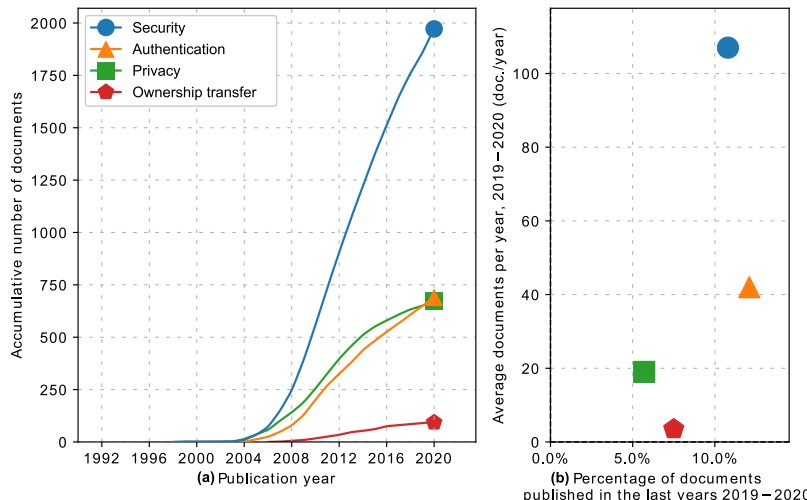

**Figure 4.** (**a**) Security and related topics, number of publications by year. (**b**) Average number of publications per year and percentage of documents published between 2019–2020.

All the expansion and diversification in Internet-related services are generating new security and privacy challenges [192]. For this reason, the most common issues, practices, and architectures are listed, to mitigate frequent privacy and security issues [193]. "Strong privacy" has been defined, and the definition has been applied to reveal vulner-

abilities in proposed privacy-enhancing RFID protocols [194]. Different security threats derived from the implementation of RFID systems in supply chain scenarios have also been explored [195]. Privacy and security have been enhanced using scalable server-less authentication and search protocols for the reader and tag communication for pervasive environments [196]. A customized infrastructure based on smart devices to enforce the privacy of individuals in healthcare scenarios has been developed [197]. Investigators are analyzing existing RFID tags regarding privacy and security aspects, providing an overview of all the types of privacy and security problems and their countermeasures [198]. In addition, the threats and measures for all the types of security threats in the IoT in general are being investigated [199].

One of the most explored topics is the mutual authentication protocol used, which has been analyzed in different ways. For example, investigations have explored possible attacks in mutual authentication protocols and proposed improvements to eliminate them [200]. Equally, there have been advances in research for new mutual agreement protocols to ensure secure communication between mobile RFID-enabled devices and the main server [201,202]. In the same way, cryptanalysis application in lightweight mutual authentication, and ownership transfer for RFID systems [203], or improvements to the hash-based RFID mutual authentication protocol to avoid denial of service (DoS) attacks have been explored [204]. Alternatively, an RFID authentication scheme based on quadratic residues has been implemented to address impersonation and replay attacks [205], and new methods such as chaotic map usage to guarantee a mutually authenticated process using a chaotic cryptosystem have been researched [206]. However, due to the tags' hardware limitations, there is a search for the most efficient ultra-lightweight authentication protocol [207]. Finally, work is in progress to confirm GS1 EPC Class 1 Generation 2 standards for RFID, including mutual authentication and privacy protection to reduce database loading and ensure user privacy [208].

Robust security and high privacy are necessary for all RFID contexts, including low-cost RFID networks [209]. All the mentioned protocols and security measures will eventually reach the general public in everyday environments. For example, deploying frameworks to preserve privacy in RFID-based healthcare systems is one of the most prominent applications for these technologies [210]. Additionally, the rapid spread of RFID makes security one of the main concerns in the context of the IoT [29].

### 4.1. Authentication

The flexibility of RFID tags in terms of the forms and variety of working frequencies allows the diverse investigation of adaptations for different and changing environments, such as solutions to guarantee secure and efficient medication for patients [211]. The definition of protocols such as the Chien's protocol and other ultra-lightweight RFID for authentication allows improvements to ensure security against known vulnerabilities [212,213]. In the same way, a lightweight authentication protocol for low-cost RFID with untraceability in mind has been described [214]. Cryptanalysis has been applied to protocols conforming to the EPC-C1G2 standard, analyzing the security level offered [215]. Similarly, the security implications in RFID and authentication processing frameworks have been analyzed, exploring a robust protocol based on error correction codes (ECC) [166], or the intrinsic usage of physical-layer authentication using integrated circuits to counteract device cloning [216]. Security has been improved by creating efficient techniques with which to monitor missing RFID tags [217]. Correspondingly, a protocol has been defined to provide coexistence proofs for RFID tags to detect tagged elements shown at the same time and in the same place [218].

### 4.2. Ownership Transfer

In enterprise-level applications and supply-chain environments, RFID tags cross different entities and places; in some cases, it is necessary to transfer control over the RFID tags, raising concerns regarding privacy when the tag travels from A to B [219]. These tasks

need to be performed securely using protocols conforming to EPCglobal class-1 generation-2 standards [220–223], and methods to improve the current protocols to offer enhanced security in ownership transfer operations are needed [224,225]. Ownership transfer is applied on mobile RFIDs, taking into consideration all the components from handheld readers to the backend server [226]. Additionally, protocols implement new methods for ensuring privacy for tag owners, and allowing secure, private, and scalable ownership-transfer [227]. Equally, it is important to ensure a secure, counterfeit-proof transaction for low-cost RFID systems where the tags have very limited resources [228,229]. To enhance the security in the transfer operation, additional measures are used, such as electronic fingerprints [230], group tag ownership transfer [231], and physically unclonable function (PUF)-based transfer for open environments using unique hardware characteristics to replace the pseudorandom generator [232]. The elliptical curve application has been used to provide controlled delegation [233]. Correspondingly, there is a search for methods with which to attack the ownership transfer system using de-synchronization based on quadratic residues [234]. However, in some cases, a protocol is needed to ensure untraceability from the previous owner in a transfer operation; to archive this, a challenge-response mechanism is proposed [235]. Moreover, new optional modes of ownership transfer are created, which let the previous owner maintain control in some specific scenarios [236]. Finally, the post-supply chain scenario has been investigated, exploring potential application in retailer and second-hand transactions or circular supply chains in the near future [237].

## 5. Discussion

Applications of RFID technologies were found in many contexts, from asset identification to tracking for industrial [5,6] and general [7] public environment applications, for instance, indoor applications [102,103], robot navigation [106,109–111], and the positioning of objects, even those not visible [104,112,117]. Ultra-wideband has been used to overcome some of the current problems with narrowband frequencies [114,118]. Additionally, it helps organizations with management and safety for construction [115,116] and pushes the limits of transmission signals' ability to enhance the quality [119–122] in different frequencies [6,118].

We found privacy and authenticity to be of utmost concern [209,228,229]. In this review, we focused on the most active RFID applications based on scientometric variables, resulting in eleven groups. The applications characterized for this technology especially concern the IoT, which supports the idea that RFID is a key requirement for the Internet of Things [28–30]. We also found RFID to be applied in supply chains [72,73], providing ways to detect counterfeiting [91,92] and keep track of any asset [90,123,127–129]. The most active security topics appear to be testing the existing protocols to search for flaws, ways to solve them [93,94,173,177,178,196,198], and balancing privacy and performance for different scenarios [192,193,196,198,199]. It was found that privacy and security are of general concern, almost in all applications. In addition, it was noted that healthcare is a topic that shows a positive AGR in the results, and it includes RFID in different related operations [8,9,149,156–163,182,197,210]. RFID is also used in access control, such as for managing access to spaces and systems [173–178], and its use in the human body has also been explored from different perspectives [189–191].

Some other recent papers have focused on specific topics, so this paper can serve as an entry point, providing a global view to help researchers to explore the most active topics, trending applications and security work in RFID, information that is valuable to people of different backgrounds and interests. During this research, it was observed that the application of RFID continues to grow, with management of the complexity in security models [34,40,62,85,177,178] and protocols [31,39,51,54–56,59–62,161,196,200–202,204,207,212,213,235].

As noted throughout the paper, RFID is used in diverse scenarios and continues to grow in its applications and functionalities, such as working with wireless sensors and IPv6, the new addressing system for the IoT, expanding its frequencies wherein technological

advances are permitted (currently to ultra-wideband). Thus, this technology can find emerging utility in RF, electronics, and security, and it is not an exaggeration to state that RFID will be the technology that will allow reaching the point, through the IoT, where every existing object will be able to be connected to the Internet.

We used the centralized and primary databases Clarivate, WoS and Scopus. Further investigations could analyze databases that harvest information rapidly to obtain cutting-edge information on RFID research activity and its relation to emerging and trending technologies.

**Author Contributions:** C.M.-A., J.R.-R., and G.R.-G. proposed the concept of this research. C.M.-A. and J.R.-R. proposed the used methodology. J.R.-R. and G.R.-G. designed the scientometric analysis. J.R.-R. do review and editing. C.M.-A. wrote the paper. G.R.-G. supervised the process. All authors have read and agreed to the published version of the manuscript.

**Funding:** This research received no external funding.

**Institutional Review Board Statement:** Not applicable.

**Informed Consent Statement:** Not applicable.

**Data Availability Statement:** Data sharing is not applicable to this article.

**Acknowledgments:** This research was conducted as part of the MSc program in Telematic Engineering at the Universidad del Cauca, Popayán, Colombia, and by the Universidad del Cauca (501100005682).

**Conflicts of Interest:** The authors declare that they have no known competing financial interests or personal relationships that have, or could be perceived to have, influenced the work reported in this article.

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
