# Peer review of "RFID Applications and Security Review"

_computation, doi:10.3390/computation9060069_

Round 1

Reviewer 1 Report

"This paper presents an interesting topic in RFID technologies application and security review. The whole papers discuss the review of different RFID applications, where the author's main concern is security review. The analysis is based on ScientoPy software to show the trending applications based on the number of papers published. Initially, As the abstract is a main section of the whole paper, I strongly recommend authors to rewrite the abstract from line 24 – 34 in page 1. I advised the authors to provide a detailed explanation of Figure 3 and Figure 4 which is missing. I am still not convinced in the discussion section. The paper tries to focus on privacy and security is a general concern. However, I advised the author to elaborate how the review can be an entry point to have a global vision as stated in line 280.

Overall, the paper has some potential to show the current RFID technologies applications and will be useful for a wide range of readers interested in the topic".

Author Response

We thank you for your time. Also, thank you for your useful and detailed review of our paper. We ordered and answered each of your points below, and in the reviewed paper version.

Please see the attachment. It contains the document's changes marking the additions in blue color and deletions in red color

1) As the abstract is a main section of the whole paper, I strongly recommend authors to rewrite the abstract from line 24 – 34 in page 1.

  • According to the reviewer's suggestion, we complemented the abstract.

2) I advised the authors to provide a detailed explanation of Figure 3 and Figure 4 which is missing.

  • According to the reviewer's suggestion, we added a detailed description of the figures.

3)  I am still not convinced in the discussion section. The paper tries to focus on privacy and security is a general concern. However, I advised the author to elaborate how the review can be an entry point to have a global vision as stated in line 280.

  • According to the reviewer's suggestion, we described in more detail this point.

Reviewer 2 Report

The paper is trying to do a review about RFID based on bibliometrics. There is some value in figures 3 and 4, but little value in listing papers as is done in almost the whole paper.
The description of referenced work is too short, vague and sometimes incomprehensible. The paper has 18 pages with 221 references, resulting in 10 pages of references for 8 pages of text. This is clearly too little text for too many references. An example is in line 88: "define a HAT:HIP address translation protocol for hybrid RFID/IP IoT communication [40]": a reader cannot understand what it means, even if the HAT and HIT acronyms had been expanded. The sentence is basically a copy of the title of paper [40]. 
Another example is in line 105-106: "Thus the effect of tag orientation and package contents [62]". The sentence means absolutely nothing, it is just part of the title of paper [62]. Another example is in line 109-110: "All this data can be important or not based on the quality of his attributes[68]". The sentence is completely vague and of no use to a reader.

My suggestion it that the authors should start by reading some RFID surveys (do a "RFID survey" query), since they should compare their review with existing surveys and a single survey ([129]) is referenced and no comparison is done. The authors should also look for the RFID papers' keywords and how frequently they are used. Using the citation impact of papers could also be useful for the analysis and it looks that it was not done at all. When data is presented, as in figures 3 and 4, it would be better to evaluate the trends over the years: are those topics becoming more referenced or less referenced? From figures 3 and 4, it is difficult to tell.  The paper does not reference standardization, and barely references technologies and protocols for RFID.

Finally, the paper is poorly written and requires an extensive English review. Only a few additional minor corrections are suggested, as there are too many to list:
- please add a space before references, e.g. "privacy [16]" instead of "privacy[16]"
- line 33, replace "First" with "first", "And" with "and"
- line 65, replace "proposes" with "purposes"
- line 66, replace "framework and protocols" with "frameworks and protocols"
- line 72, sentence is missing a subject. I suggest something like "Papers equally analyse how to enforce security..."
- line 75, replace "The RFID" with "RFID"
- line 80, line 82, line 87, who is "they" referring to?
- line 85 is exceeding the right margin
- line 98, replace "studding" with "studying"
- line 114, replace "critic" with "critical"

Summarizing, the paper needs to be considerably improved before it can be accepted.

Author Response

We thank you for your time. Also, thank you for your useful and detailed review of our paper. We ordered and answered each of your points below, and in the reviewed paper version.

Please see the attachment. It contains the document's changes marking the additions in blue color and deletions in red color.

1)  There is some value in figures 3 and 4,

  • According to the reviewer's suggestion, we added a detailed description of the figures.

2)  but little value in listing papers as is done in almost the whole paper.   The description of referenced work is too short, vague and sometimes incomprehensible. The paper has 18 pages with 221 references, resulting in 10 pages of references for 8 pages of text. This is clearly too little text for too many references. An example is in line 88: "define a HAT:HIP address translation protocol for hybrid RFID/IP IoT communication [40]": a reader cannot understand what it means, even if the HAT and HIT acronyms had been expanded. The sentence is basically a copy of the title of paper [40]. Another example is in line 105-106: "Thus the effect of tag orientation and package contents [62]". The sentence means absolutely nothing, it is just part of the title of paper [62]. Another example is in line 109-110: "All this data can be important or not based on the quality of his attributes[68]". The sentence is completely vague and of no use to a reader.

  • According to the reviewer's suggestion, we improved the description of referenced work.

3)  My suggestion it that the authors should start by reading some RFID surveys (do a "RFID survey" query) since they should compare their review with existing surveys and a single survey ([129]) is referenced and no comparison is done.

  • According to the reviewer's suggestion, we added the surveys and reviews and make a comparison.

4)  The authors should also look for the RFID papers' keywords and how frequently they are used.

  • According to the reviewer's suggestion, we clarify this in the description of the figures.

5) Using the citation impact of papers could also be useful for the analysis and it looks that it was not done at all.

  • This process has been done to select the referenced papers using the combined dataset obtained from the tool ScyentoPy.

6) When data is presented, as in figures 3 and 4, it would be better to evaluate the trends over the years: are those topics becoming more referenced or less referenced? From figures 3 and 4, it is difficult to tell.  

  • According to the reviewer's suggestion, we clarify this in the description of the figures.

7)  The paper does not reference standardization, and barely references technologies and protocols for RFID.

  • As we describe, the topics are selected by their activity and citation number, resulting in the topics and papers referenced in the work.

8) Finally, the paper is poorly written and requires an extensive English review. Only a few additional minor corrections are suggested, as there are too many to list: 
- please add a space before references, e.g. "privacy [16]" instead of "privacy[16]"
    - line 33, replace "First" with "first", "And" with "and"
    - line 65, replace "proposes" with "purposes"
    - line 66, replace "framework and protocols" with "frameworks and protocols"
    - line 72, sentence is missing a subject. I suggest something like "Papers equally analyse how to     enforce security..."
    - line 75, replace "The RFID" with "RFID"
    - line 80, line 82, line 87, who is "they" referring to?
    - line 85 is exceeding the right margin
    - line 98, replace "studding" with "studying"
    - line 114, replace "critic" with "critical"

  • According to the reviewer's suggestion, we correct these points and similar ones.

Round 2

Reviewer 1 Report

I found authors have done effort in reshaping the research article. However, I have few concerns:

  1. I am still not clear from line 84 – 95. Authors need to demonstrate Figure 3 in terms of comparison and analysis of those dataset instead of writing the same data sets in terms of sentences. Same for Figure 4 from line 280-285.
  2. It seems the authors are directly using the references paper titles. I have never seen such a way of writing research articles which includes a whole title of reference papers. For example: Line 218-219, 291-292, 330-331, 209-210
  3. It seems figure 1 caption needs to be rewritten if its author's original contribution otherwise needs to be cited.
  4. The use of 237 research articles as a reference demonstrate that the authors want to produce a good research article. However, I am still not convinced with most of the sections of the review paper including the discussion section.

Author Response

We thank you for your time. Also, thank you for your useful and detailed review of our paper. We ordered and answered each of your points below, and in the reviewed paper version.

For clarity, the comments are included in the section below listed with our comment to each issue listed in between with red color.

------------------------------

I found authors have done effort in reshaping the research article. However, I have few concerns:

  • I am still not clear from line 84 – 95. Authors need to demonstrate Figure 3 in terms of comparison and analysis of those dataset instead of writing the same data sets in terms of sentences. Same for Figure 4 from line 280-285.
    • According to the suggestion, we fixed the section

  • It seems the authors are directly using the references paper titles. I have never seen such a way of writing research articles which includes a whole title of reference papers. For example:

Line 218-219, 291-292, 330-331 209-210

    • According to the suggestion, a profound revision was conducted.
  • It seems figure 1 caption needs to be rewritten if its author's original contribution otherwise needs to be cited.
    • According to the suggestion, the caption was revised
  • The use of 237 research articles as a reference demonstrate that the authors want to produce a good research article. However, I am still not convinced with most of the sections of the review paper including the discussion section.
    • According to the suggestion, the sections were reviewed in detail

Reviewer 2 Report

The paper has been slightly improved over the previous version, but still needs to be significantly improved to be acceptable for publication.
Some surveys were added in the 3rd paragraph of the introduction, but the comparison is very poor and is missing many surveys. If you want to get some recent surveys, you can search in Google Scholar, specifying a year range. For instance for RFID surveys since 2015, you could search:
https://scholar.google.com/scholar?q=RFID+survey&hl=en&as_sdt=0%2C5&as_ylo=2015&as_yhi=

Figures 3 and 4 were not modified at all. It would be better to evaluate the trends over the years: are those topics becoming more referenced or less referenced? Are they topics that emerged recently, or old topics? From figures 3 and 4, it is difficult to tell. If you have the data, why don't you plot the data for each year?  The abstract says "We present fundamental indicators such as (...) average growth rate, and the average documents per year". However, those indicators are not presented nor discussed in the paper. 

The paper presents a very superficial review of the RFID topic, mostly counting applications and security aspects and missing technologies, standards and protocols. The paper title includes the word "technologies" that are not properly addressed throughout the paper. Which technologies, standards and protocols are emerging, stable or losing interest? Which ones are more widely used?

The paper is poorly written and still requires an extensive English review. Only a few additional minor corrections for the first page are suggested, as there are too many to list:
- the paper title is missing a comma between "technologies" and "applications"
- lines 9-10, "Selecting from the results the following sections" makes no sense, as sections do not come from the results.
- line 15, "letting" makes no sense. May be "presenting" is what it should read.
- line 34, replace "currier" with "courier"
- line 35, why is "Its" with a capital letter? The sentence has no verb.

Author Response

We thank you for your time. Also, thank you for your useful and detailed review of our paper. We ordered and answered each of your points below, and in the reviewed paper version.

For clarity, the comments are included in the section below listed with our comment to each issue listed in between with red color.

------------------------------

The paper has been slightly improved over the previous version, but still needs to be significantly improved to be acceptable for publication.

  • Some surveys were added in the 3rd paragraph of the introduction, but the comparison is very poor and is missing many surveys.
    • For this paper, we keep the most relevant reviews and surveys, according to the title change resulted from the recommendation in point five.
  • If you want to get some recent surveys, you can search in Google Scholar, specifying a year range. For instance for RFID surveys since 2015, you could search: https://scholar.google.com/scholar?q=RFID+survey&hl=en&as_sdt=0%2C5&as_ylo=2015&as_yhi=
    • The main data sources for this paper are limited to WoS and Scopus.
  • Figures 3 and 4 were not modified at all. It would be better to evaluate the trends over the years: are those topics becoming more referenced or less referenced? Are they topics that emerged recently, or old topics? From figures 3 and 4, it is difficult to tell.
    • According to the reviewer's suggestion, the graphs were updated.
  • If you have the data, why don't you plot the data for each year?  The abstract says "We present fundamental indicators such as (...) average growth rate, and the average documents per year". However, those indicators are not presented nor discussed in the paper.
    • According to the reviewer's suggestion, we updated the tables containing the described metrics.
  • The paper presents a very superficial review of the RFID topic, mostly counting applications and security aspects and missing technologies, standards and protocols. The paper title includes the word "technologies" that are not properly addressed throughout the paper.
    • According to the reviewer's suggestion, we updated the title.

Which technologies, standards and protocols are emerging, stable or losing interest? Which ones are more widely used?

    • This information can be now consulted in the new graphics and tables, using the average growth (AGR) rate and publications per year (ADY, ADLY). 
  • The paper is poorly written and still requires an extensive English review. Only a few additional minor corrections for the first page are suggested, as there are too many to list

- the paper title is missing a comma between "technologies" and "applications"

lines 9-10, "Selecting from the results the following sections" makes no sense, as sections do not come from the results.
- line 15, "letting" makes no sense. May be "presenting" is what it should read.
- line 34, replace "currier" with "courier"
- line 35, why is "Its" with a capital letter? The sentence has no verb.

    • According to the reviewer's suggestion, we conducted an English review with external help.

Round 3

Reviewer 1 Report

I found authors have made effort in reshaping the research article. Address the issue of all my two previous reviews. The modified version is able to demonstrate the review paper based on RFID applications and security.

Author Response

We thank you for your time. Also, thank you for your useful and detailed review of our paper. We have done a deeper English review making various corrections in the manuscript.

Changes in the manuscript are shown in blue for additions, and in red strike-through for removed sentences, in the differential PDF version.

Reviewer 2 Report

Figures 3 and 4 were updated to reflect the evolution of the RFID topics over the years. Tables 2 and 3 were added with bibliographic indicators. Most of the proposed corrections were properly done. The quality of the English text was also improved, although it still requires a revision.

Thus, it is this reviewer opinion that the paper can be considered for publication after a deeper English review.

Only a few additional minor corrections are suggested, as there are too many to list:
- line 20, replace "As for RFID, is implemented" with "As RFID is implemented"
- line 94, replace "en the Figure 3" with "in Figure 3"
- line 279, replace "augmenting adding" with "adding"
- line 440, replace "As we encounter analyzing" with "Analyzing"
- line 445, replace "RFID continue" with "RFID continues"
- line 446, replace "[114,118] Although" with "[114,118]. Although" (add a full stop)
- line 446-447, replace "it helps (...) and push" with "it helps (...) and pushes"
- line 456, replace "keep track on" with "keep track of"
- line 473, replace "security models. [34,40,62,85,177,178] and protocols" with "security models [34,40,62,85,177,178] and protocols" (remove full stop).

Author Response

We thank you for your time. Also, thank you for your useful and detailed review of our paper. We have done a deeper English review making various corrections in the manuscript, addressing your recommendations, and finding additional ones.

Changes in the manuscript are shown in blue for additions, and in red strike-through for removed sentences, in the differential PDF version.
